# Dynamics in the Sakaguchi-Kuramoto model with bimodal frequency distribution

Shuangjian Guo[1], Yuan Xie[2], Qionglin Dai[1], Haihong Li[1], Junzhong Yang[1]*

**1** School of Science, Beijing University of Posts and Telecommunications, Beijing, People's Republic of China, **2** Faculty of Science, Xi'an Aeronautical University, Xi'an, People's Republic of China

* jzyang@bupt.edu.cn

**Data Availability Statement:** All relevant data are within the paper.

**Funding:** The author(s) received no specific funding for this work.

## Abstract

In this work, we study the Sakaguchi-Kuramoto model with natural frequency following a bimodal distribution. By using Ott-Antonsen ansatz, we reduce the globally coupled phase oscillators to low dimensional coupled ordinary differential equations. For symmetrical bimodal frequency distribution, we analyze the stabilities of the incoherent state and different partial synchronous states. Different types of bifurcations are identified and the effect of the phase lag on the dynamics is investigated. For asymmetrical bimodal frequency distribution, we observe the revival of the incoherent state, and then the conditions for the revival are specified.

## Introduction

Collective behaviors emerged out of a large number of interacting units are common in nature. As one type of collective behavior characterizing the phase coherence in nonidentical units, synchronization is well recognized in various systems such as fireflies flashing in unison [1, 2], applauding persons in a large audience [3], pedestrians [4, 5], and others [6]. Kuramoto model (KM) is the paradigmatic model in the field of synchronization [7, 8]. There are two key simplifications in the original KM, which renders the analytical treatments to be possible. Firstly, each unit is treated as a phase oscillator, which is valid for the weak coupling situation where the amplitude information of each unit is inessential to the collective behaviors. The dynamics of each phase oscillator is solely determined by its natural frequency and in turn the frequencies of all oscillators are drawn from a prescribed frequency distribution function $g(\omega)$. Secondly, the coupling between units is assumed to be a global one and takes the form of a sinusoidal function with the same strength $K$. The coupling strength together with the frequency distribution determine the dynamics of KM.

Previously, KM has been intensively investigated. A variety of its generalizations have been proposed and many interesting phenomena have been observed. The repulsive interaction among oscillators ($K < 0$) may be introduced to KM. Tsimring et al. [9] found that KM with repulsive interaction fails to synchronize. Hong and Strogatz [10, 11] treated the coupling strength as an oscillator's ability to response to the mean field and found $\pi$ synchronous states and novel time-dependent traveling wave synchronous states in the presence of both repulsive and attractive interaction. Yuan et al. further considered the $\pi$ synchronous state in the presence of correlation between the conformists/contrarians and the natural frequencies of

**Competing interests:** The authors have declared that no competing interests exist.

oscillators [12]. Zhang et al. [13] introduced frequency-weighted coupling to KM and found explosive synchronization and chimera-like states. KM has also been extended to complex networks where network topology can affect the synchronization transition. In [14], the authors assigned the natural frequencies of phase oscillators to be the degrees of the nodes they locate on network and found explosive synchronization transition. Recently, KM with higher-order interaction such as biharmonic interaction has drawn some attentions in which infinitely many stable partial synchronous states and a continuum of abrupt desynchronization transition have been identified [15]. The shear is a crucial nonlinear ingredient for complex behaviors in coupled systems [16]. Time delay was also investigated [17–19], and small time delay can be approximated by a phase lag parameter $\beta$. Along this line, the the phase lag $\beta$ is introduced into the coupling function as $K\sin(\theta_j - \theta_i + \beta)$ so that KM is generalized to Sakaguchi-Kuramoto model (SKM) [20] and the synchronous dynamics has been investigated [16, 21–24].

Actually, the original KM is concise enough to display rich dynamics by taking proper frequency distribution $g(\omega)$. It has been theoretically shown that the transition to synchronization occurs at $K_c = 2/[\pi g(0)]$ [25] for even and unimodal $g(\omega)$. Above $K_c$, the incoherent state yields to a stationary partial synchronous state. For asymmetrical unimodal $g(\omega)$, the partial synchronous states are always time-dependent [26]. When $g(\omega)$ becomes a bimodal one, increasing coupling strength always first leads to a standing wave state, in which two synchronous clusters of oscillators oscillate at opposite mean frequencies and, then, to traveling wave states, in which synchronous oscillators rotate at the same frequency [27]. Bimodal frequency distributions in the KM were already investigated at different levels [28–34], and trimodal frequency distribution were also studied [30]. Under proper parameters, KM with bimodal distribution gives rise to discontinuous transitions cross different dynamical states. Martens et al. [33] studied KM with bimodal natural frequency distribution consisting of two equally weighted Lorentzians, and they derived the system's stability diagram. They found three states depending on the parameters and initial conditions, incoherent state, partial synchronous state, and standing wave synchronous states. They also presented analytical results for the bifurcation boundaries between these states. Omel'chenko and colleagues [35] studied SKM with $g(\omega)$ being a superposition of two unimodal frequency distributions with the same mean frequency. They found a nonuniversal synchronization transition in which the incoherent state may be revived at stronger coupling strength after it yields to partial synchronous state at $K_c$. Asymmetry has also been studied recently [26, 29, 31]. For more complicate frequency distribution such as a trimodal one, KM may display collective chaos through a cascade of period-doubling bifurcations [36].

In this work, we study SKM with bimodal natural frequencies distribution. As a natural extension of Ref. [33], the phase-lag parameter $\beta$ is introduced into the model. The paper is organized as follows. In section 2, we present the model and reduce the coupled phase oscillators to a low-dimensional coupled ordinary differential equations. In section 3, we first study the synchronous dynamics in the model with symmetrical bimodal frequency distribution with an emphasis on the effects of the phase lag. Different dynamical states are analyzed and different types of bifurcations are identified. Then we consider SKM with asymmetrical bimodal frequency distribution. We study the revival phenomenon of the incoherent state and investigate the dependence of revival of the incoherent state on parameters. Summary is made in the last section.

## Materials and methods

We consider N phase oscillators with global coupling and the motion equation follows

$$\dot{\theta}_i = \omega_i + \frac{K}{N}\sum_{j=1}^{N} \sin(\theta_j - \theta_i - \beta), \qquad (1)$$

with $\theta_i$ the phase of oscillator $i$ and $K$ is the global coupling strength. $\beta$ is the phase lag parameter resulting in rich interesting dynamical phenomena and the model reduces to the original KM at $\beta = 0$. $\omega_i$ is the natural frequency of oscillator $i$, which is chosen randomly from a probability distribution $g(\omega)$. In this work, we assume that the frequency distribution $g(\omega)$ takes the form

$$g(\omega) = \frac{1}{\pi} \left[ \frac{p_1 \Delta_1}{(\omega - \omega_1)^2 + \Delta_1^2} + \frac{p_2 \Delta_2}{(\omega - \omega_2)^2 + \Delta_2^2} \right] \tag{2}$$

with $p_1 + p_2 = 1$ and $\omega_1 = -\omega_2 = \omega_0$. The parameters $\Delta_{1,2}$ measure the heterogeneity of oscillators in their natural frequencies. Generally, both the heterogeneity parameter $\Delta$ and the phase lag $\beta$ have strong effects on the synchronous dynamics. However, these two parameters impact on the collective dynamics in different way. $\Delta$ is used to measure the fraction of oscillators to be in synchronization. Large $\Delta$ always suggests small fraction of phase oscillators to be in synchronization. In contrast, $\beta$ measures the phase mismatch between the synchronous oscillators and the mean field. Sufficiently large $\beta$ pushes synchronous phase oscillators to be in antiphase with the mean field, which downgrades synchronization and tends to destroy the coherence in population. Recent work points out that incoherent state may be revival at proper choice of $\beta$ [35], which suggests the non-monotonic effects of $\beta$ on the coherence in population.

The synchronous dynamics in the model (1) is measured by the complex order parameter, defined as $Z = R e^{i\Theta} = \frac{1}{N} \Sigma_j e^{i\theta_j}$. $|Z| = 0$ suggests the incoherent state and, otherwise, a synchronous state. Using the order parameter, Eq (1) is reformulated as

$$\dot{\theta}_i = \omega_i - KR \sin(\theta_i - \Theta + \beta). \tag{3}$$

To study the dynamics, we consider the thermodynamic limit ($N \to \infty$) where Eq (1) can be written in a continuous formulation in terms of a probability density $f(\theta, \omega, t)$, defined as the fraction of oscillators with natural frequency between $\omega$ and $\omega + d\omega$ and phase between $\theta$ and $\theta + d\theta$ at time $t$, which satisfies the normalization condition $\int_{-\infty}^{\infty} \int_0^{2\pi} f(\theta, \omega, t) d\theta d\omega = 1$ and $\int_0^{2\pi} f(\theta, \omega, t) d\theta = g(\omega)$. The probability density evolves following the continuity equation

$$\frac{\partial f}{\partial t} + \frac{\partial (fv)}{\partial \theta} = 0 \tag{4}$$

with

$$v = \omega + \frac{iK}{2} [Z^* e^{i(\theta + \beta)} - Z e^{-i(\theta + \beta)}]. \tag{5}$$

The order parameter $Z$ in the continuous formalism is reformulated as

$$Z(t) = \int_{-\infty}^{\infty} \int_0^{2\pi} f(\theta, \omega, t) e^{i\theta} d\theta d\omega. \tag{6}$$

Since the probability density is periodic in $\theta$, it can be expanded in Fourier series as

$$f(\theta, \omega, t) = \frac{g(\omega)}{2\pi} \left[ 1 + \sum_{n=1}^{\infty} f_n(\theta, \omega, t) e^{in\theta} + c.c. \right] \tag{7}$$

with $c.c.$ the complex conjugate of the previous term. Ott and Antonsen proposed an ansatz (OA ansatz) [37] that the coefficients $f_n(\omega, t)$ obey $f_n(\omega, t) = [\alpha(\omega, t)]^n$. Substituting Eq (4) with

the ansatz, we obtain

$$\frac{\partial \alpha}{\partial t} + \frac{K}{2}\left(Ze^{-i\beta}\alpha^2 - Z^*e^{i\beta}\right) + i\omega\alpha = 0 \tag{8}$$

with

$$Z^*(t) = \int_{-\infty}^{\infty} g(\omega)\alpha(\omega, t)d\omega. \tag{9}$$

For the natural frequency distribution Eq (2), the order parameter $Z$ becomes

$$Z(t) = p_1 z_1(t) + p_2 z_2(t), \tag{10}$$

where we denote $z_{1,2}(t) = \alpha^*(\omega_{1,2} - i\Delta_{1,2}, t)$. Then the synchronization in the model (1) is characterized by the sub-order parameters $z_{1,2}(t)$. The evolution of $z_i$ ($i = 1, 2$) follows

$$\dot{z}_i = -(\Delta_i - i\omega_i)z_i + \frac{K}{2}\left(Ze^{-i\beta} - Z^*e^{i\beta}z_i^2\right). \tag{11}$$

Furthermore, we let $\alpha_j = z_j^* = r_j e^{-i\phi_j}$ ($j = 1, 2$) and introduce $\psi = \phi_1 - \phi_2$. Then substituting them into Eq (11), we have

$$\begin{aligned}
\dot{r}_1 &= -\Delta_1 r_1 + \frac{K}{2}(1 - r_1^2)[p_1 r_1 \cos\beta + p_2 r_2 \cos(\beta + \psi)], \\
\dot{r}_2 &= -\Delta_2 r_2 + \frac{K}{2}(1 - r_2^2)[p_2 r_2 \cos\beta + p_1 r_1 \cos(\psi - \beta)], \\
\dot{\psi} &= \omega_1 - \omega_2 - \frac{K(r_1^2 + 1)}{2}\left[\frac{p_2 r_2 \sin(\beta + \psi)}{r_1} + p_1 \sin\beta\right] \\
&\quad + \frac{K(r_2^2 + 1)}{2}\left[\frac{p_1 r_1 \sin(\beta - \psi)}{r_2} + p_2 \sin\beta\right].
\end{aligned} \tag{12}$$

The presence of the phase lag in the model breaks the symmetry between $r_1$ and $r_2$ even when $\Delta_1 = \Delta_2$ and $p_1 = p_2$. Eq (12) consisting of three coupled ordinary differential equations is equivalent to the model (1, 2) and, therefore, the dynamics of the model (1, 2) may be reflected by $r_1$, $r_2$, and $\psi$. To be mentioned, the partial synchronous states in the model (1) [or the reduced model (12)] are always time-dependent, periodic or quasiperiodic, for nonzero $\beta$. In the reduced model (12), these time-dependent synchronous states are reduced to equilibria or periodic solutions by considering the model in a rotating frame characterizing the time-dependent $\phi_1$. In the following, we claim a solution to be an equilibrium or periodic one according to its behavior in the reduced model (12).

## Results and discussion

### Symmetric frequency distribution

We first consider the symmetric frequency distribution where $p_1 = p_2 = 0.5$, and $\Delta_1 = \Delta_2 = \Delta$. We set the coupling strength $K = 4$ and investigate the effect of the phase lag $\beta$ on the model dynamics.

We start with the reduced model Eq (11) and investigate the stability of the incoherent state. The incoherent state is defined by $z_1 = z_2 = 0$. Supposing that the evolution of perturbations to the incoherent state follows $\delta z_{1,2} \sim e^{\lambda t}$ and substituting them into Eq (11), we may

have

$$\lambda_{1,2} = e^{i\beta} - \Delta' \pm \sqrt{e^{2i\beta} - \omega_0'^2} \tag{13}$$

with $\Delta' = 4\Delta/K$ and $\omega_0' = 4\omega_0/K$. For convenience, we assume $Re(\lambda_1) > Re(\lambda_2)$. When $Re(\lambda_1)$ becomes positive, the incoherent state becomes unstable. Beyond the bifurcation, Eq (12) gives rise to two new stable equilibria except for the unstable incoherent state, $r_{1,2} > 0$ in one equilibrium, and $r_{1,2} < 0$ in the other which is unrealistic and should be discarded. Therefore, the incoherent state undergoes a supercritical Pitchfork bifurcation when $Re(\lambda_1)$ crosses zero (we denoted it as PB1). Interestingly, when $Re(\lambda_2)$ crosses zero, it induces another pitchfork bifurcation (denoted as PB2) in which two newborn equilibria are unstable and one of them is unrealistic. The pitchfork bifurcations involving the incoherent state occur at the critical curves described by

$$\Delta' = \cos\beta \pm \frac{\sqrt{2}}{2}\sqrt{\cos(2\beta) - \omega_0'^2 + \sqrt{1 + \omega_0'^4 - 2\omega_0'^2 \cos(2\beta)}}. \tag{14}$$

When $\beta = 0$, the critical curves (14) are reduced to a semicircle $\Delta' = 1 \pm \sqrt{1 - \omega_0'^2}$ for $\omega_0' < 1$, which is related to pitchfork bifurcation, and a line $\Delta' = 1$ for $\omega_0' > 1$ which is related to Hopf bifurcation [33]. Increasing $\beta$ from zero, the stability regime of the incoherent state shrinks in the plane of $\Delta'$ and $\omega_0'$.

Then we consider model dynamics by focusing on Eq (12). The equilibria to Eq (12) represent the partial synchronous states and their stabilities can be analyzed by the linear stability method. For $\beta = 0$, the partial synchronous state can be acquired rigorously by setting $r_1 = r_2$ [33]. However, for partial synchronous state, $r_1 = r_2$ is always not held as $\beta \neq 0$. the equilibria to Eq (12) are obtained by numerical methods and their stabilities are determined by the eigenvalues of the Jacobian matrices at them. To illustrate, we consider the bifurcation diagrams along three parameter paths by setting $\beta = 0.1$ and $K = 4$. Firstly, we consider the parameter path with $\omega_0'$ from 0.4 to 2 at $\Delta' = 0.4$. The bifurcation diagrams are presented in Fig 1(a) where $r_1$ and $r_2$ are plotted, respectively. Besides the incoherent state which is always unstable along this parameter path, there are at most four equilibria denoted as $FP_i^{(1)}$ ($i = 1, 2, 3, 4$). The eigenvalues of the corresponding Jacobian matrices at these equilibria are plotted in Fig 1(b) and 1(c). As shown, the equilibria $FP_1^{(1)}$ is stable until $\omega_0' \simeq 1.62$ at which it collides with a saddle $FP_2^{(1)}$ and gives rise to a limit cycle, a standing wave synchronous state, through a SNIPER bifurcation (saddle node infinite period bifurcation). The equilibrium $FP_2^{(1)}$ is a saddle with a one-dimensional unstable manifold, which is born at $\omega_0' \simeq 1.40$ with another saddle $FP_3^1$ owning a two-dimensional unstable manifold through a saddle-node bifurcation (denoted as SN2). Shortly after SN2, the unstable $FP_3^{(1)}$ is turned into a saddle-focus. The equilibrium $FP_4^{(1)}$ has a pair of complex conjugate eigenvalues whose real parts are positive and is an unstable saddle-focus, which is produced by the pitchfork bifurcation (denoted as PB2) of the incoherence state at around $\omega_0' \simeq 0.8$ according to Eq (14). Along this parameter path, there are two stable synchronous states, one is represented by $FP_1^1$ before $\omega_0' = 1.62$ and the other is represented by a limit cycle [the solid curves in Fig 1(a)].

Secondly, we consider the parameter path with $\omega_0'$ from 1.15 to 1.22 at $\Delta' = 0.95$. The bifurcation diagram is presented in Fig 2(a) and the eigenvalues for all equilibria are presented in Fig 2(c) and 2(e). As shown, we find two stable equilibria ($FP_1^{(2)}, FP_2^{(2)}$) and one unstable equilibrium ($FP_3^{(2)}$). The bifurcations at which $FP_1^{(2)}$ and $FP_3^{(2)}$ annihilate with each other and at which $FP_2^{(2)}$ and $FP_3^{(2)}$ are born in pair belong to the saddle-node bifurcation (one is denoted as

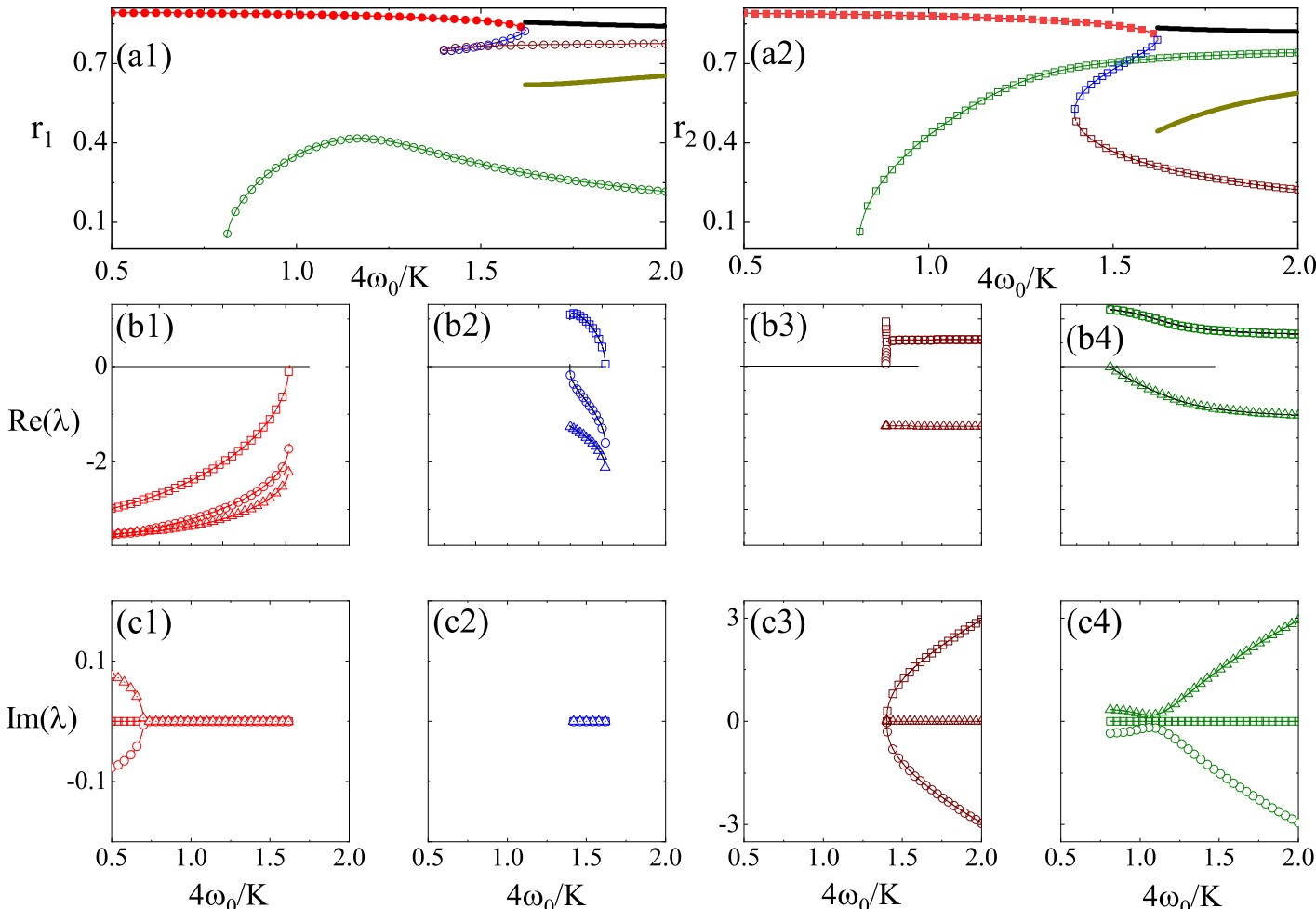

**Fig 1. (Color online) Bifurcation diagrams of $r_1$, $r_2$ and $\lambda$ against $\omega' = 4\omega_0/K$.** $K = 4$, $\beta = 0.1$, and $\Delta' = 4\Delta/K = 0.4$. Solid (open) data points represent stable (unstable) states. In top panels, red, blue, wine, and dark green symbols are for partial synchronous states $FP_1^{(1)}$, $FP_2^{(1)}$, $FP_3^{(1)}$, and $FP_4^{(1)}$, respectively. Thick black and dark green lines represent the maximum and minimum values of $r_1$ and $r_2$ for stable standing wave synchronous states. In middle and bottom panels, from left to right, real and imaginary parts of the eigenvalues $\lambda$ for partial synchronous states from $FP_1^{(1)}$ to $FP_4^{(1)}$ are displayed. Squares, circles, and triangles denote eigenvalues $\lambda_1$, $\lambda_2$, and $\lambda_3$, respectively.

SN1 and the other is SN2). Along this path, the bistability between $FP_1^{(2)}$ and $FP_2^{(2)}$ exists in a range of $\omega_0'$.

The third parameter path is chosen against $\Delta'$ at $\omega_0' = 1.5$, which is presented in Fig 2(b), 2(d) and 2(f). There are two equilibria, $FP_{1,2}^{(3)}$, and a stable periodic solution. $FP_1^{(3)}$ is a focus, which changes from an unstable to a stable one by colliding with the limit cycle at $\Delta' \simeq 0.904$ through a Hopf bifurcation (denoted as HB). Furthermore, the stable $FP_1^{(3)}$ disappears at $\Delta' \simeq 1.09$ by turning the unstable incoherent state to being stable one through a pitchfork bifurcation (PB1). The unstable equilibrium $FP_2^{(3)}$ is always unstable, which results from a pitchfork bifurcation (PB2) of the unstable incoherent state when the real part of its second eigenvalue $Re(\lambda_2)$ crosses zero [see Eq (13)].

Using the above analysis, the phase diagrams in the plane of $\Delta'$ and $\omega_0'$ at $\beta = 0$ and $\beta = 0.1$ are presented in Fig 3(a) and 3(b), respectively. Actually, the results at $\beta = 0$ have been thoroughly explored [33] and there is only two minor modifications in Fig 3(a). Firstly, we point

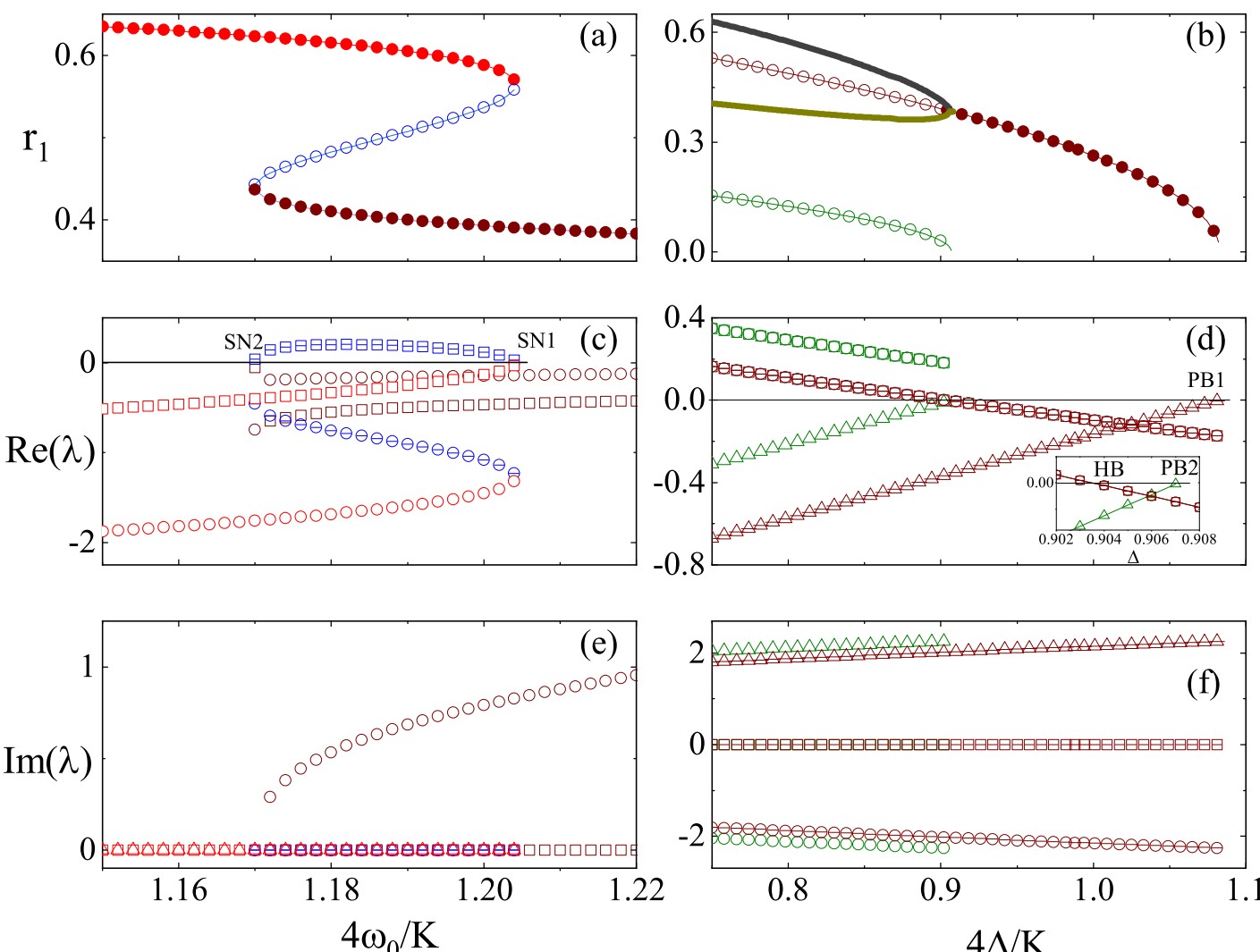

**Fig 2. (Color online) Bifurcation diagrams of $r_1$ and $\lambda$ against $\omega' = 4\omega_0/K$ at $\Delta' = 4\Delta/K = 0.95$ (left column) and against $\Delta'$ at $\omega' = 1.5$ (right column).** Solid (open) data points represent stable (unstable) states. In (a), red, blue, and wine lines are for partial synchronous states $FP_1^{(2)}$, $FP_2^{(2)}$, and $FP_3^{(2)}$, respectively. In (b), wine and dark green lines are for partial synchronous states $FP_1^{(3)}$ and $FP_2^{(3)}$, respectively. Thick black and dark green lines refer to the standing wave synchronous state. In the panels from (c) to (f), squares, circles, and triangles denote the real and imaginary parts of eigenvalues $\lambda_1$, $\lambda_2$, and $\lambda_3$, respectively. The inset of (d) shows that HB occurs at a lower $\Delta'$ than PB2. Note that the incoherent state changes its stability across the pitchfork bifurcation (PB1).

out that the incoherent state loses its stability through a pitchfork bifurcation at low $\omega_0'$ instead of a transcritical bifurcation claimed in Refs. [33, 38], which is similar to Refs. [39]. We find that the bifurcations are similar to those in KM with trimodal frequency distribution [30]. Secondly, we include in the phase diagram one more saddle-node bifurcation (SN2) which involves the birth of a pair of unstable saddles. The saddles arising from SN2 were not reported in Ref. [33] in which the authors concerns with the long-term dynamics at $\beta = 0$. Interestingly, we find that one of these two saddles becomes stable as $\beta \neq 0$. To be stressed, unstable solutions have no effects on the long-term dynamics of the model dynamics. However, the existence of unstable solutions greatly shapes the topological structure of the underlying phase space and has strong impacts on the transient dynamics of the model. Moreover, under certain conditions, unstable solutions might become stable with the change of parameter and, then,

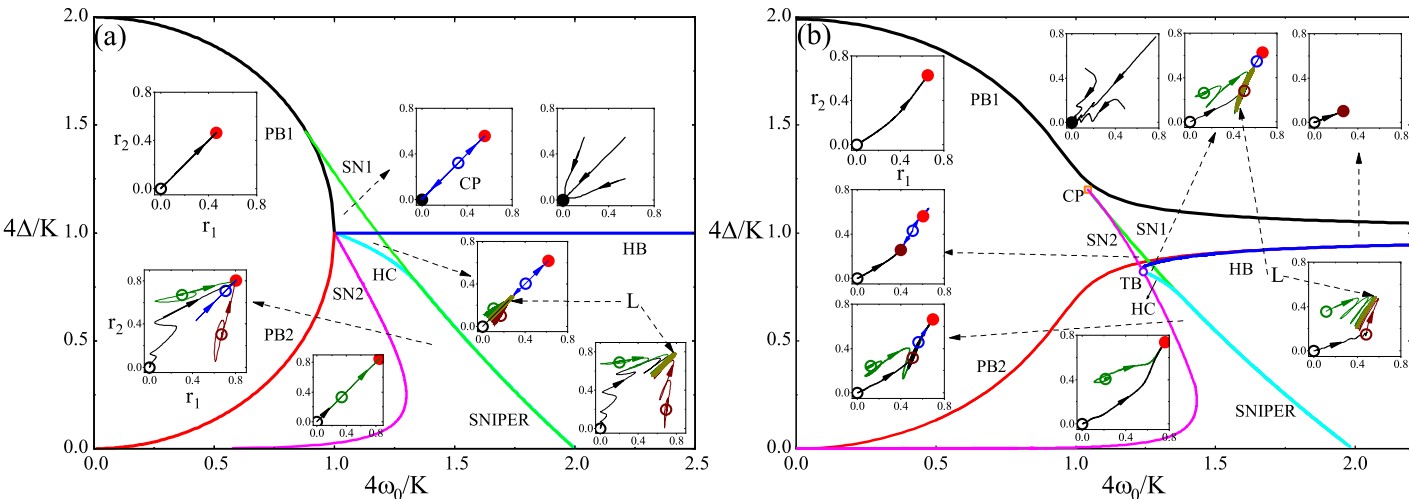

**Fig 3. (Color online) Bifurcation diagrams on the ($\Delta' = 4\Delta/K$, $\omega' = 4\omega_0/K$) plane for (a) $\beta = 0$ and (b) $\beta = 0.1$.** Line color codes: black and red for two pitchfork bifurcations, PB1 and PB2, respectively; green and pink for two saddle-node bifurcations, SN1 and SN2, respectively; blue for HB (Hopf bifurcation); cyan for HC (homoclinic bifurcation). Acronyms: SNIPER for saddle node infinite period; CP for cusp point of SN1 and SN2; TB for Takens-Bogdanov point. To present the topological structure of the phase space in different phase domains, we plot the phase portraits on the ($r_1$, $r_2$) plane in several insets with the parameters chosen from different phase domains. The dashed arrows pointing to insets refer to the phase domain represented by the insets. In each inset, several phase portraits (wiggly lines) are plotted with arrows representing the evolution from or towards the solutions in Eq (12). In these insets, solid (open) dots represent stable (unstable) partial synchronous states, while the dark yellow curves represent stable standing wave partial synchronous state denoted by $L$. The solutions in the same color in different insets are the same solution. $K = 4$.

take effects on the long-term dynamics of the model. Therefore, in the perspective of stability analysis, the exploration of unstable solutions is still necessary.

In Fig 3(b), there are two pitchfork bifurcations involving the incoherence (PB1,PB2), two saddle-node bifurcations involving partial synchronous states (SN1,SN2), and three bifurcations involving limit cycle synchronous states (Hopf bifurcation, homoclinic bifurcation, and SNIPER). The critical curves relating to these bifurcations divide the parameter plane of $\Delta/4K$ and $\omega_0/4K$ into several domains. And the phase diagram in Fig 3(b) shows that $FP_1^{(1)}$ and $FP_1^{(2)}$ are the same type of solutions while $FP_4^{(1)}$ and $FP_2^{(3)}$ are the same type of solutions. The typical evolutions on the plane of $r_1$ and $r_2$ from (or towards) the solutions in these different domains are presented in the insets.

Compared with Fig 3(a), there are several unique features in Fig 3(b) to be addressed. At $\beta = 0$, the two PBs form a continuous semicircle. However, these two PBs become two separated curves. Furthermore, the Hopf bifurcation underlies the transition between the stable incoherent state and the stable limit cycle at $\beta = 0$. However, at nonzero $\beta$, the Hopf bifurcation occurs between the stable partial synchronous state and the stable limit cycle. In addition, the Hopf bifurcation stays much close to PB2 of the incoherent state. At $\beta = 0$, there exists a domain in which the incoherent state coexists with a partial synchronous state. However, no coexistence between the incoherent state and any partial synchronous states at $\beta = 0.1$, as shown in Fig 3(b). Instead, there exists the coexistence between two partial synchronous states in the domain enclosed by two saddle-node bifurcations (SN1 and SN2) and HB. As shown in Fig 3(b), there exists a Takens-Bogdanov bifurcation (denoted as TB) where Hopf bifurcation, homoclinic bifurcation (denoted as HC), and saddle-node bifurcation merge. Interestingly, a pair of stable and unstable synchronous states are born at SN2 above TB while a pair of unstable synchronous states occur at SN2 below TB. In addition, SN1 gradually merges with HC to become SNIPER.

In KM with unimodal frequency distribution, increasing the phase lag $\beta$ always downgrades the coherence in population and, when $\beta = \pi/2$, the critical coupling strength $K$ for the onset of

synchronization becomes infinite. However, in SKM with nonzero $\beta$, the phase lag $\beta$ impacts on the coherence in population in a non-monotonic way. To see it clearly, we consider several slices on the parameter plane of $\Delta'$ and $\omega_0'$ and vary $\beta$. Fig 4 shows the phase diagrams on the plane of $\Delta'$ and $\beta$ at different $\omega_0'$. We find that, at large $\omega_0'$, the incoherent state first becomes unstable and, then, regains its stability again with the increase of $\beta$ from zero to $\pi/2$ [see Fig 4 (c) and 4(d)] though increasing $\beta$ always favors the stability of the incoherent state at small $\omega_0'$. Fig 4 also tells us that, at small $\omega_0'$, the incoherent states always yields to a partial synchronous state through PB [see Fig 4(a) and 4(b)] while both partial synchronous states and standing wave synchronous state may appear at high $\omega_0'$ [Fig 4(d)]. For intermediate $\omega_0'$ such as $\omega_0' = 1.25$ in Fig 4(c), complicated structure in the phase diagram appears, for example the bistability between different partial synchronous states, the bistability between the partial synchronous states and the standing wave states, and the existence of two Takens-Bagdanov bifurcations.

### Asymmetric frequency distribution

Omel'chenko and colleagues have found an interesting phenomenon in a SKM with $g(\omega)$ being a superposition of two unimodal frequency distributions with the same mean frequency where the incoherent state may be revived at stronger coupling strength [35]. Liu and colleagues found the same phenomenon in a SKM with $g(\omega_0)$ being the superposition of two bimodal frequency distributions [40].

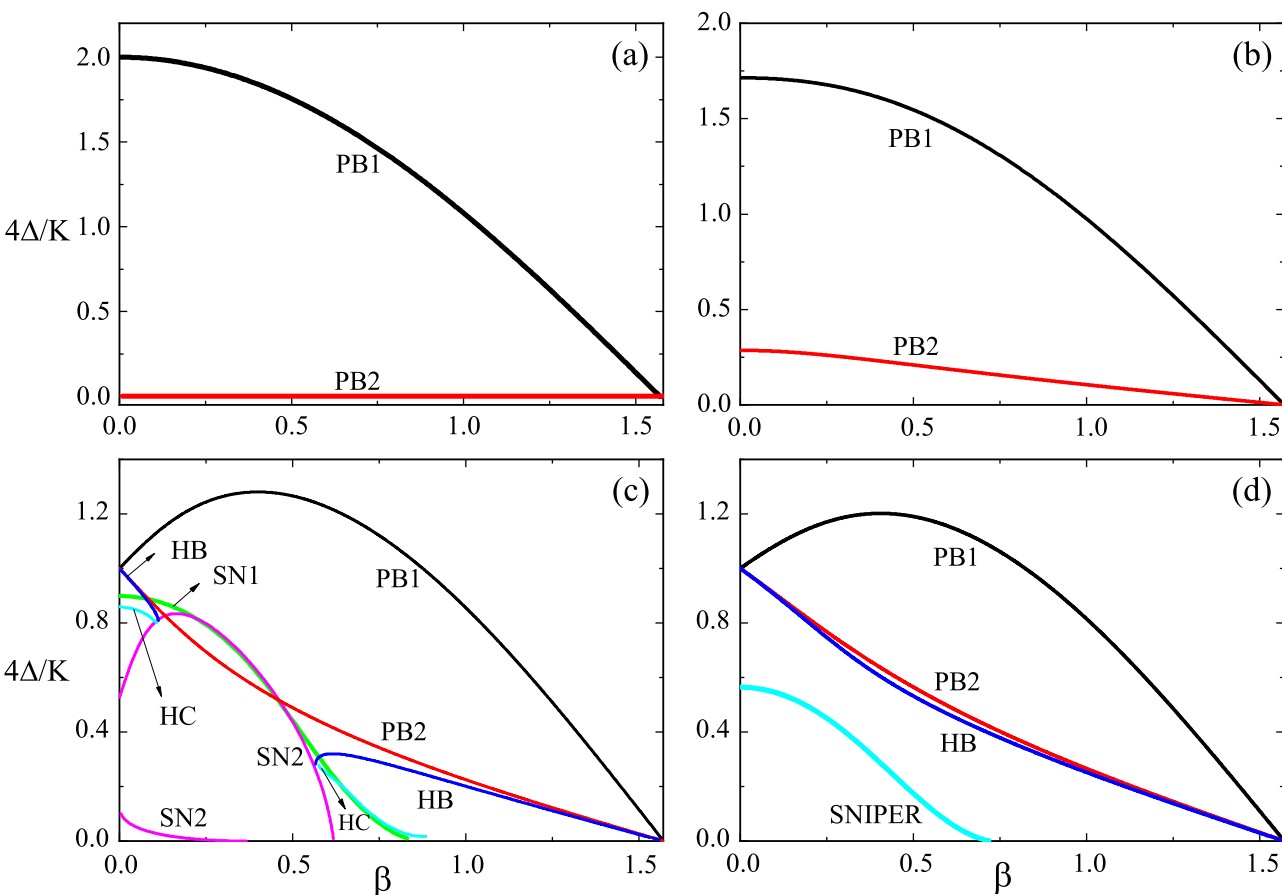

**Fig 4. (Color online) Phase diagrams on the plane of $\Delta' = 4\Delta/K$ and $\beta$ at $\omega_0' = 4\omega_0/K = 0$ in (a), $\omega_0' = 0.7$ in (b), $\omega_0' = 1.25$ in (c), and $\omega_0' = 1.5$ in (d). $K = 4$.**

Here we show the revival of the incoherent state for asymmetrical bimodal frequency distribution and provide the conditions for better observing revival of the incoherent state. We consider the stability diagrams of the incoherent state on different parameter planes where the stability of the incoherent state is calculated based on Eq (11). With reference to the process of reaching Eq (13), we may have

$$Re(\lambda_{1,2}) = \frac{K}{4}\cos\beta - \frac{\Delta_1 + \Delta_2}{2} \pm \frac{1}{4\sqrt{2}}\sqrt{c_1 + \sqrt{c_1^2 + c_2^2}}$$

$$c_1 = 4\Delta_-^2 - 4\omega_-^2 - 4Kp_-\Delta_-\cos\beta + 4Kp_-\omega_-\sin\beta + K^2\cos 2\beta$$

$$c_2 = -K^2\sin 2\beta - 8\Delta_-\omega_- + 4Kp_-\omega_-\cos\beta + 4Kp_-\Delta_-\sin\beta$$

$$(15)$$

with $\Delta_- = \Delta_1 - \Delta_2$, $\omega_- = \omega_1 - \omega_2 = 2\omega_0$ and $p_- = p_1 - p_2$. To be mentioned, $\omega_0$ may be negative when the peak frequency $\omega_1$ is less than the peak frequency $\omega_2$. Positive and negative $\omega_0$ may exert different impacts on the model dynamics due to the asymmetrical bimodal frequency distribution. Incoherent states change stability with changing parameters at hopf bifurcation or pitchfork bifurcation [25, 26, 29, 31, 35]. We can get the bifurcation curves in Fig 5 from Eq (15), in which the more general conditions are considered analytically. Fig 5(a) shows the

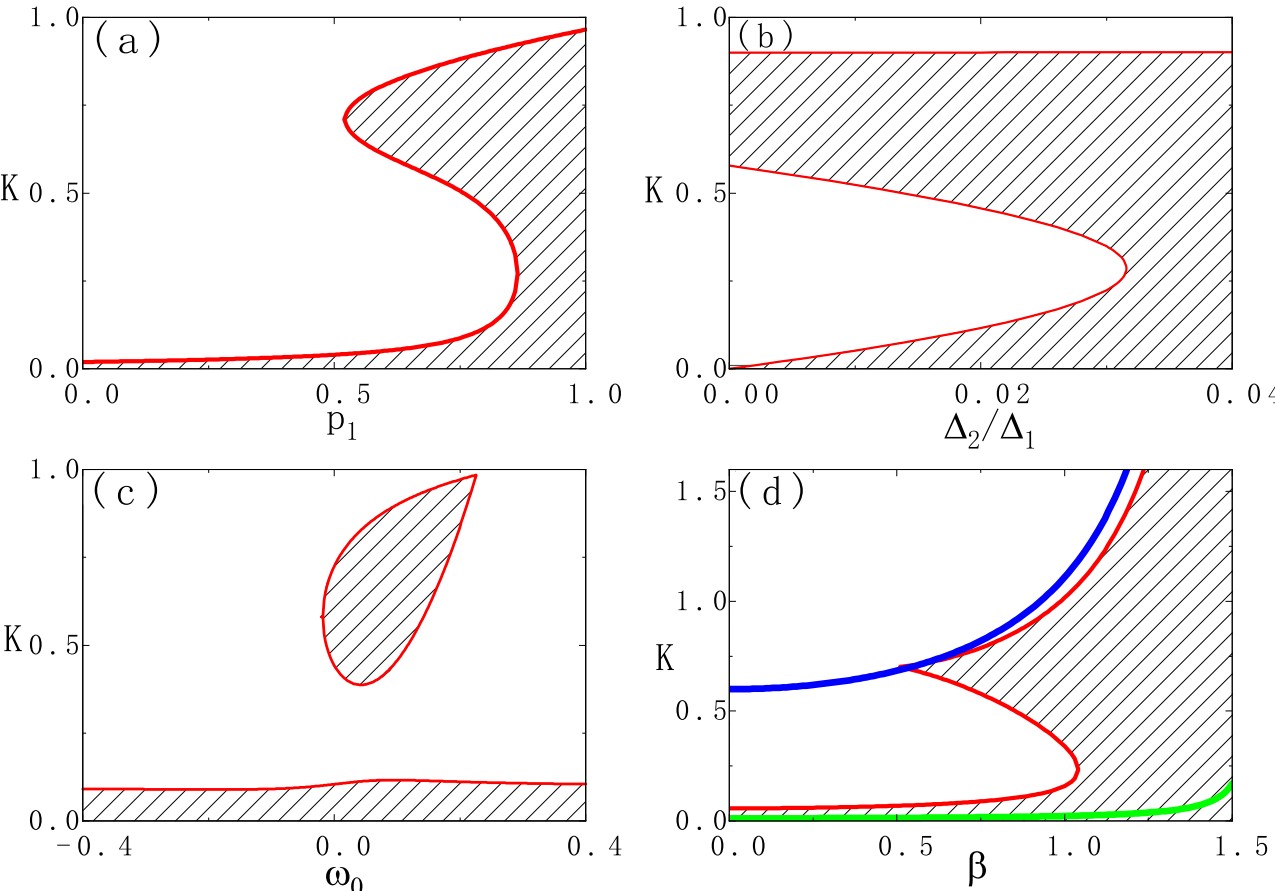

**Fig 5. (Color online) Stability diagrams of the incoherent state for asymmetrical bimodal frequency distribution on various parameter planes.** (a) $(K, p_1)$ plane at $\beta = 0.9$, $\Delta_1 = 0.3$, $\Delta_2/\Delta_1 = 0.02$, and $\omega_0 = 0.1$; (b) $(K, \Delta_2/\Delta_1)$ plane at $\beta = 0.9$, $\Delta_1 = 0.3$, $\omega_0 = 0.1$, and $p_1 = 0.8$; (c) $(K, \omega_0)$ plane at $\beta = 0.9$, $\Delta_1 = 0.3$, $\Delta_2/\Delta_1 = 0.02$, and $p_1 = 0.8$; (d) $(K, \beta)$ plane at $\Delta_1 = 0.3$, $\Delta_2/\Delta_1 = 0.02$, $\omega_0 = 0.1$, and $p_1 = 0.8$. The shaded regions with red boundary lines, obtained from Eq (15), mark the stable incoherent state. The blue and green lines in (d) are critical curves $K = 2\Delta_1/\cos\beta$ for $p_1 = 1$ and $K = 2\Delta_2/\cos\beta$ for $p_1 = 0$, respectively.

results on the plane of $K$ and $p_1$. For $p_1 = 0$ ($K_c \simeq 0.019$), the incoherent state becomes unstable at sufficient weak coupling strength while strong coupling strength is required for $p_1 = 1$ ($K_c \simeq$ 0.965), which can be seen from Eq (11). Between these two extreme situation, there exists a domain at around $p \in (0.52, 0.86)$ in which the revival of the incoherent state appears. The stability diagram on the plane of $K$ and $\Delta_2/\Delta_1$ with fixed $\Delta_1$ shows that the revival of the incoherent state requires sufficiently small $\Delta_2/\Delta_1$ and it becomes the most prominent at $\Delta_2/\Delta_1 = 0$ [see Fig 5(b)]. If we measure the revival phenomenon of the incoherent state by the range of the coupling strength $K$, Fig 5(c) indicates that the superposition of two unimodal distributions with the same mean frequency is not the best candidate for realization of the revival phenomenon. Weak mismatch between the center frequencies of the two unimodal distribution is the optimal for the revival of the incoherent state. Finally, Fig 5(d) suggests that revival of the incoherent state occurs only in SKM with proper phase lag $\beta$. We also plot the critical curves, $K = 2\Delta_1/\cos\beta$ and $K = 2\Delta_2/\cos\beta$, for the incoherent state when $p_1 = 1$ and $p_1 = 0$. It is interesting to find that these two curves may approximate part of the boundary of the stable incoherent state, which suggests that the revival of the incoherent state is somehow induced by the competition between these two instability mechanisms. To summarize, the revival of the incoherent state studied here requires some conditions. Firstly, the frequency distribution is composed of two unimodal ones and the sufficiently low ratio of their widths is required for the revival of the incoherent state. Secondly, that the fraction of oscillators with the natural frequency from the fat peak in the population is higher than that from the thin peak is required for the revival of the incoherent state. Thirdly, proper choice of $\beta$ is required. These conditions are similar to those reported in the previous work [35]. Different from the work [35] where the two unimodal distributions share the same central frequency and the frequency distribution is a symmetrical one, the frequency distribution here is a bimodal one and no symmetry on it is required. The results in Fig 5 suggest that the revival of the incoherent state could be a rather popular phenomenon.

## Conclusion

In conclusion, we have investigated the globally coupled Sakaguchi-Kuramoto model with bimodal natural frequency distributions. By using Ott-Antonsen ansatz for dimension reduction, we reduce the coupled phase oscillators to a low dimensional coupled ordinary equations. For symmetrical bimodal frequency distribution, we analyze the linear stabilities of the incoherent state and partial synchronous states and identify different types of bifurcations between different dynamical states. Especially, the impacts of the phase lag $\beta$ on the model dynamics are studied. For example, nonzero $\beta$ greatly modifies the topological structure of the phase space and unfolds certain bifurcations degenerated at $\beta = 0$. More importantly, $\beta$ impacts on synchronous dynamics in the population in a non-monotonic way. The bifurcation may be unfolded by nonzero. We also study the revival of the incoherent state for the model with asymmetrical bimodal frequency distributions and the conditions for better observing the phenomenon are proposed.

## Author Contributions

**Conceptualization:** Shuangjian Guo, Qionglin Dai, Junzhong Yang.

**Data curation:** Shuangjian Guo, Yuan Xie, Qionglin Dai, Junzhong Yang.

**Formal analysis:** Shuangjian Guo, Yuan Xie, Qionglin Dai, Haihong Li, Junzhong Yang.

**Funding acquisition:** Qionglin Dai, Junzhong Yang.

**Investigation:** Shuangjian Guo, Yuan Xie, Qionglin Dai, Junzhong Yang.

**Methodology:** Shuangjian Guo, Yuan Xie, Qionglin Dai, Haihong Li, Junzhong Yang.

**Project administration:** Qionglin Dai, Haihong Li, Junzhong Yang.

**Resources:** Shuangjian Guo, Qionglin Dai, Junzhong Yang.

**Software:** Shuangjian Guo, Yuan Xie, Qionglin Dai, Haihong Li, Junzhong Yang.

**Supervision:** Shuangjian Guo, Yuan Xie, Haihong Li, Junzhong Yang.

**Validation:** Shuangjian Guo, Yuan Xie, Qionglin Dai, Haihong Li, Junzhong Yang.

**Visualization:** Shuangjian Guo, Yuan Xie, Haihong Li, Junzhong Yang.

**Writing – original draft:** Shuangjian Guo, Qionglin Dai, Junzhong Yang.

**Writing – review & editing:** Shuangjian Guo, Qionglin Dai, Junzhong Yang.

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
