## [Decision Letter · Decision Letter 0]

29 Jul 2020

PONE-D-20-12498

Dynamics in the Sakaguchi-Kuramoto Model with bimodal frequency distribution

PLOS ONE

Dear Dr. Guo,

Thank you for submitting your manuscript to PLOS ONE. After careful consideration, we feel that it has merit but does not fully meet PLOS ONE’s publication criteria as it currently stands. Therefore, we invite you to submit a revised version of the manuscript that addresses the points raised during the review process.

We look forward to receiving your revised manuscript.

Kind regards,

Per Sebastian Skardal

Academic Editor

PLOS ONE

Additional Editor Comments:

Dear Dr. Gao,

First, thank you for submitting your work to PLOS Ones and thank you for your patience through this difficult period. The review of your work has been delayed due to a number of Referees inability to submit reviews in these turbulent times. Ultimately, we have decided to proceed initially with the one report that we currently have on hand.

We invite you to resubmit your work once you have responded to each of the Referee's comments, provided below.

Best,

Per Sebastian Skardal

Journal Requirements:

"This work is supported by National Natural Science Foundation of China under Grants

No. 11575036 and No. 11805021, and BUPT Excellent Ph.D. Students Foundation

under Grant No. CX2019138."

Reviewers' comments:

Reviewer's Responses to Questions

**Comments to the Author**

1. Is the manuscript technically sound, and do the data support the conclusions?

Reviewer #1: Partly

2. Has the statistical analysis been performed appropriately and rigorously? 

Reviewer #1: N/A

3. Have the authors made all data underlying the findings in their manuscript fully available?

Reviewer #1: Yes

4. Is the manuscript presented in an intelligible fashion and written in standard English?

Reviewer #1: No

5. Review Comments to the Author

Reviewer #1: The authors study and report the bifurcation structure of the Sakaguchi-Kuramoto model with bimodal distribution. The work is a natural extension of Martens et al., PRE 2009 that allows for asymmetric solutions and includes a phase shift parameter \\beta.

The first set of results are obtained by increasing \\beta from zero, and examining the nature of the equilibrium and limit cycle solutions. In particular, the changes that occur in the two-dimensional bifurcation diagram are presented.

The second set of results involve the "revival" of the incoherent state. The authors consider various ways to break the symmetry of the bimodal natural frequency distribution. They find parameter space regions where the stability of the incoherent state goes from stable to unstable to stable again by varying one parameter.

In general, the work appears to be largely correct and interesting. However, there are a few problems with citations and terminology, and the exposition of the results section should be improved.

First, this work is so closely related to Martens et al. (2009) -- Ref. 35 in the manuscript -- that it should be placed explicitly in that context. Specifically, Martens et al. (2009) should be introduced and cited by name in the Introduction of this manuscript (the way that many other references are cited there). The results of Martens et al. (2009) should be discussed, and the relationship of the authors' work to that reference described.

Second, the authors refer to pitchfork bifurcations of the incoherent state (i.e., the origin). Pitchfork bifurcations involve one equilibrium transitioning into three with a change of stability of the original equilibrium. In the system studied, the origin (which has pairs of degenerate eigenvalues) transitions into two equilibria. The authors mention that the apparently missing equilibrium corresponds to an "unrealistic" solution with a negative values of the absolute values of z1 and z2.

The authors also claim that other publications (their refs. 35 and 36) that have identified these bifurcations of the incoherent state as (degenerate) transcritical bifurcations are wrong. If the authors want to say this, then they need to establish their claim by providing the relevant details.

If I remember correctly, one way to view the bifurcation of the incoherent state is to consider two coincident equilibria existing at the origin (due to the degeneracy). At the transcritical bifurcation, these exchange stability and the stable one one migrates away from the origin.

Have the authors examined the leading-order nonlinearity? For the normal forms of the transcritical and pitchfork bifurcations, these are quadratic and cubic, respectively.

Third, the readability of the manuscript would be vastly improved if several large paragraphs were split into smaller paragraphs as I suggest below. I also include in the following list several other corrections.

Please find a logical way to break the second paragraph of the introduction (lines 29-68) into smaller paragraphs.

Line 141: Please confirm if you want "always not" or "not always" here. These have very different meanings.

Line 150: till  until

Line 156: conjugated complex  complex conjugate

Line 162-3: the reference to the panels b and c in Fig. 2 is incorrect.

The paragraph on page 7 is hard to read. It would be very helpful for the reader if this text is split into multiple paragraphs as follows:

Line 161: begin a new paragraph with "Secondly, we consider...".

Line 168: begin a new paragraph with "The third parameter...".

Line 175: begin a new paragraph with "Using [the] above analysis, ...". Insert "the" as shown.

Line 179: Join this text to the (new) preceding paragraph.

In the first line of the caption of Figure 1, it would be helpful if "4\\omega_0/K" were replaced with "\\omega_0'=4\\omega_0/K". This would help the reader connect the discussion in the text, which uses \\omega_0', to the Figure. Please make similar adjustments to the other figures.

Lines 182-183: Regarding the second point mentioned here, the unstable saddles that arise from curve SN2 have r1 != r2. That is why it was not reported in Martens et al. (2009); they only considered solutions with r1 = r2. Please clarify this point.

Line 184: Start a new paragraph at "In Fig 3(b)".

Are the wiggly lines in the insets in Fig. 3 sample trajectories? These are confusing.

Line 189: The sentence that begins "Now it is clear...": This claim is not clear from looking at Figure 3.

Line 192: New paragraph at "Compared with...".

The contradictory statements in lines 198-200 are confusing. What are you trying to say? Please reword.

Similarly, the sentences in lines 210-212 contradict each other. I don't understand what the authors want the reader to see in Figure 4.

Figure 5 caption: change "sparse shadow region" to "shaded region".

6. PLOS authors have the option to publish the peer review history of their article (what does this mean?). If published, this will include your full peer review and any attached files.

Reviewer #1: No

---

## [Author Response · Author response to Decision Letter 0]

14 Sep 2020

Manuscript No.: PONE-D-20-12498

Title: Dynamics in the Sakaguchi-Kuramoto Model with bimodal frequency distribution

Dear Editor,

Thank you very much for handling our manuscript. We really appreciate the insightful comments from the Referee. We have revised the manuscript based on these comments. We hope this revised version could meet the publication standard of PLOS ONE.

For the revised submission, we would like to update our Funding Statement as follows: “This work is supported by National Natural Science Foundation of China under Grants No. 11575036 and No. 11805021, and BUPT Excellent Ph.D. Students Foundation under Grant No. CX2019138. The funders had no role in study design, data collection and analysis, decision to publish, or preparation of the manuscript.”

In the initial submission, we provided the funding statement in the manuscript but did not state them in the Funding section of the online submission. We must apologize for our carelessness. We would be grateful if you could help us update the financial disclosure statement.

The followings are the details of the reply. 

Reviewer #1: The authors study and report the bifurcation structure of the Sakaguchi-Kuramoto model with bimodal distribution. The work is a natural extension of Martens et al., PRE 2009 that allows for asymmetric solutions and includes a phase shift parameter \\beta.

The first set of results are obtained by increasing \\beta from zero, and examining the nature of the equilibrium and limit cycle solutions. In particular, the changes that occur in the two-dimensional bifurcation diagram are presented.

The second set of results involve the "revival" of the incoherent state. The authors consider various ways to break the symmetry of the bimodal natural frequency distribution. They find parameter space regions where the stability of the incoherent state goes from stable to unstable to stable again by varying one parameter.

In general, the work appears to be largely correct and interesting. However, there are a few problems with citations and terminology, and the exposition of the results section should be improved.

Comment 1: First, this work is so closely related to Martens et al. (2009) -- Ref. 35 in the manuscript -- that it should be placed explicitly in that context. Specifically, Martens et al. (2009) should be introduced and cited by name in the Introduction of this manuscript (the way that many other references are cited there). The results of Martens et al. (2009) should be discussed, and the relationship of the authors' work to that reference described. 

Reply: Thank you very much for the suggestions. We have mentioned the work of Martens et al. (2009) in the introduction section and added several sentences to discuss the results of them (lines 50-54). We also clarified the relationship of our work to the reference in the main text (lines 61-62, 180-182). 

Comment 2: Second, the authors refer to pitchfork bifurcations of the incoherent state (i.e., the origin). Pitchfork bifurcations involve one equilibrium transitioning into three with a change of stability of the original equilibrium. In the system studied, the origin (which has pairs of degenerate eigenvalues) transitions into two equilibria. The authors mention that the apparently missing equilibrium corresponds to an "unrealistic" solution with a negative values of the absolute values of z1 and z2.

The authors also claim that other publications (their refs. 35 and 36) that have identified these bifurcations of the incoherent state as (degenerate) transcritical bifurcations are wrong. If the authors want to say this, then they need to establish their claim by providing the relevant details. If I remember correctly, one way to view the bifurcation of the incoherent state is to consider two coincident equilibria existing at the origin (due to the degeneracy). At the transcritical bifurcation, these exchange stability and the stable one migrates away from the origin. Have the authors examined the leading-order nonlinearity? For the normal forms of the transcritical and pitchfork bifurcations, these are quadratic and cubic, respectively.

Reply: Thanks for the comments. The incoherent state does undergo a supercritical Pitchfork bifurcation when $ Re(\\lambda_1)$ crosses zero. Since solving Eq. (12) gives rise to two new stable equilibria except for the unstable incoherent state at the bifurcation, $r_{1,2}>0$ in one equilibrium, and $r_{1,2}<0$ in the other. Due to the definition, $r_{1,2}<0$ is unrealistic though Eq.(12) allows for it. In Figure R_1 attached following, we show the equilibria acquired from Eq. (12) against $4\\Delta/K $, which supports our claim.

(Attached Figure R_1 is viewing in “Response to Reviewers.doc”.)

Figure R_1: Bifurcation diagrams of $r_2$ (a) and $r_1$ (b) against $\\Delta’=4\\Delta/K$ obtained from Eq. (12). The partial synchronous states ($FP^{(3)}_1$ in wine and $FP^{(3)}_2$ in dark green) and incoherent state (in black) are stable (unstable) states denoted by solid (open) symbols. To support our claim that they are not transcritical bifurcations but pitchfork bifurcations, the unrealistic equilibria ($r_{1,2}<0$) are also shown. Across the pitchfork bifurcation (PB1), the incoherent state changes the stability. And the $FP^{(3)}_1$ ($FP^{(3)}_2$) disappear at the pitchfork bifurcation PB1(PB2) with increasing $\\Delta’=4\\Delta/K$. The parameters, $\\beta=0.1$ and $\\omega'_0=4\\omega_0/K=1.5$, are same to Fig. 2(b) in the main text.

Comment 3: Third, the readability of the manuscript would be vastly improved if several large paragraphs were split into smaller paragraphs as I suggest below. I also include in the following list several other corrections.

-Please find a logical way to break the second paragraph of the introduction (lines 29-68) into smaller paragraphs.

-Line 141: Please confirm if you want "always not" or "not always" here. These have very different meanings.

-Line 150: till  until

-Line 156: conjugated complex  complex conjugate

-Line 162-3: the reference to the panels b and c in Fig. 2 is incorrect.

-The paragraph on page 7 is hard to read. It would be very helpful for the reader if this text is split into multiple paragraphs as follows:

Line 161: begin a new paragraph with "Secondly, we consider...".

Line 168: begin a new paragraph with "The third parameter...".

Line 175: begin a new paragraph with "Using [the] above analysis, ...". Insert "the" as shown.

Line 179: Join this text to the (new) preceding paragraph.

-In the first line of the caption of Figure 1, it would be helpful if "4\\omega_0/K" were replaced with "\\omega_0'=4\\omega_0/K". This would help the reader connect the discussion in the text, which uses \\omega_0', to the Figure. Please make similar adjustments to the other figures.

Reply: Thank you very much for these constructive comments. We have modified the manuscript by following these suggestions.

-Lines 182-183: Regarding the second point mentioned here, the unstable saddles that arise from curve SN2 have r1 != r2. That is why it was not reported in Martens et al. (2009); they only considered solutions with r1 = r2. Please clarify this point.

Reply: Thank you for the suggestion. We have clarified this point in the manuscript (lines 180-182).

-Line 184: Start a new paragraph at "In Fig 3(b)".

-Are the wiggly lines in the insets in Fig. 3 sample trajectories? These are confusing.

Reply：Yes, they are sample trajectories which show the trajectories from the same colored unstable fixed points. To make this point clear, we have added one sentence in the caption of Fig. 3 to clarify it. 

-Line 189: The sentence that begins "Now it is clear...": This claim is not clear from looking at Figure 3.

-Line 192: New paragraph at "Compared with...".

-The contradictory statements in lines 198-200 are confusing. What are you trying to say? Please reword.

-Similarly, the sentences in lines 210-212 contradict each other. I don't understand what the authors want the reader to see in Figure 4.

-Figure 5 caption: change "sparse shadow region" to "shaded region".

Reply：We have reorganized these sentences or words to make them clear in the revised manuscript (for examples, lines 198-200, lines 211-212). Thank you very much for pointing them out.

We appreciate the help of Professor Mei Zhang from Department of Physics, Beijing Normal University. She helps us improve the language of the manuscript. We hope the English writing of the revised manuscript could meet the journal requirements.

---

## [Decision Letter · Decision Letter 1]

14 Oct 2020

PONE-D-20-12498R1

Dynamics in the Sakaguchi-Kuramoto Model with bimodal frequency distribution

PLOS ONE

Dear Dr. Guo,

Thank you for submitting your manuscript to PLOS ONE. After careful consideration, we feel that it has merit but does not fully meet PLOS ONE’s publication criteria as it currently stands. Therefore, we invite you to submit a revised version of the manuscript that addresses the points raised during the review process. In particular, please attend to Reviewer #2's comments. 

We look forward to receiving your revised manuscript.

Kind regards,

Per Sebastian Skardal

Academic Editor

PLOS ONE

Reviewers' comments:

Reviewer's Responses to Questions

**Comments to the Author**

1. If the authors have adequately addressed your comments raised in a previous round of review and you feel that this manuscript is now acceptable for publication, you may indicate that here to bypass the “Comments to the Author” section, enter your conflict of interest statement in the “Confidential to Editor” section, and submit your "Accept" recommendation.

Reviewer #1: All comments have been addressed

Reviewer #2: (No Response)

2. Is the manuscript technically sound, and do the data support the conclusions?

Reviewer #1: Yes

Reviewer #2: Yes

3. Has the statistical analysis been performed appropriately and rigorously? 

Reviewer #1: N/A

Reviewer #2: Yes

4. Have the authors made all data underlying the findings in their manuscript fully available?

Reviewer #1: Yes

Reviewer #2: Yes

5. Is the manuscript presented in an intelligible fashion and written in standard English?

Reviewer #1: Yes

Reviewer #2: Yes

6. Review Comments to the Author

Reviewer #1: The authors have addressed my concerns. Ambiguities have been removed and the English is now sufficiently intelligible.

Reviewer #2: The authors of the manuscript “Dynamics in the Sakaguchi-Kuramoto Model with bimodal frequency distribution” investigate the dynamics of a population of phase oscillators whose natural frequency terms are drawn from a bimodal distribution. The coupling between oscillators is global and occurs via the sine of their pairwise phase differences. A phase lag parameter, or frustration parameter, is added, which leads to a bimodal Kuramoto-Sakaguchi model. The model in this particular form and with respect to asymmetric frequency distributions has not been analyzed in the literature, yet. Moreover, the authors report the “revival” of the incoherent state: when increasing a bifurcation parameter, the stable incoherent state first becomes unstable and later on stabilizes again. The collective behavior of the system is studied following the Ott-Antonsen ansatz, which allows for deriving analytically tractable low-dimensional dynamics of the Kuramoto order parameter. Here, the system consists of three coupled ordinary differential equations, whose bifurcation structure is investigated with analytic and numerical techniques.

The model is well introduced by putting it into a larger context, and also by referring it to directly related literature, which makes it interesting to the readership of PLOS ONE. Although the here-reported form of the Kuramoto-Skaguchi model is new, the mathematically sound results for the symmetric case, which is central to this manuscript, can be anticipated from Martens, Bick and Panaggio (Chaos 26, 094819, 2016) and Refs. 20, 29, when taking into account that there is an equivalence between the bimodal formulation and a two-coupled-population formulation of coupled phase oscillators as established, e.g., in Ref. 30. The manuscript can present an important additional contribution to the field if, in a revised version, the authors can explicate whether the phase lag parameter \\beta and the heterogeneity parameter \\Delta have differential effects on the collective dynamics.

The novelty of this manuscript is the impact of the asymmetric bimodal distribution on the revival of the incoherent state, which unfortunately is not yet worked out thoroughly. Since similar revival results have been obtained by Omelchenko and Wolfrum in Ref. 33 for unimodal frequency distributions, the manuscript can significantly be improved when the authors discuss in more detail the differences and similarities with their work and that by Omelchenko and Wolfrum.

If these two concerns above can adequately be addressed in a revised version, I can happily support publication of the manuscript in PLOS ONE.

Further points that should be addressed in a revised version of the manuscript:

1) As to the presentation of results, the authors should clarify why unstable solutions and their bifurcations (as in Figs. 1, 2, 3 and 4) are studied extensively. As their numerical analysis attains large attention through Figs. 1 and 2, the reader is misled in that unstable solutions seem to be crucial. But I doubt that they have any real effect on the dynamics of the population activity [apart from the SN2 curve in Fig. 3b between the CP and TB points].

2) The presentation of the resulting dynamical regimes appears unclear: why are the sub-order parameters r_1 and r_2 the important properties? The insets in Fig. 3 suggest that r_1 and r_2 really are the central properties, but why they are of interest, does not become clear. And why is not the global order parameter Z characteristic for the network dynamics?

3) The insets in Fig. 3 are hard to understand. A more intuitive explanation, and/or schematics of the global states, would be helpful for the reader. Maybe the layout of inset figures can be changed to that used in Bick, Martens and Panaggio (Chaos 2016, Chaos 2018).

4) I don’t understand what Fig. 4 adds to the manuscript. If the purpose is to highlight how beta changes the bifurcation diagram, then there are at least two better options:

a. Add more bifurcation diagrams similar to Fig. 3 for different choices of beta.

b. Make Fig. 3b 3-dimensional by adding a beta-axis.

5) Line 220: Please start a new subsection in order to distinguish the parts using a symmetric versus an asymmetric distribution, respectively.

6) In Fig. 5c, \\omega_0 takes on negative values, but since it is considered to be half the distance between the peaks of the bimodal frequency distribution, how can it be negative? Please clarify

7) In the Conclusion, line 257, please enumerate here your main findings in which way (how) beta affects the dynamics. Moreover, it would helpful to show whether beta has any differential effects compared to the other important parameter \\Delta because, in general, both prevent the system to fully synchronize. In line 211 the authors already point to the counter-intuitive phenomenon that close to 4\\Delta = K, a larger frustration parameter \\beta “tend[s] to destabilize the incoherent state”, but the mechanism why this is so remains unclear.

8) Line 259: what are these conditions for better observing the revival? Please write them out here explicitly and, perhaps, the authors can also elaborate here on their hypothesis in lines 248/249

7. PLOS authors have the option to publish the peer review history of their article (what does this mean?). If published, this will include your full peer review and any attached files.

Reviewer #1: No

Reviewer #2: No

---

## [Author Response · Author response to Decision Letter 1]

11 Nov 2020

Manuscript No.: PONE-D-20-12498R2

Title: Dynamics in the Sakaguchi-Kuramoto Model with bimodal frequency distribution

Dear Editor,

Thank you very much for handling our manuscript. We really appreciate the insightful comments from the Referee. We have revised the manuscript based on these comments. We hope this revised version could meet the publication standard of PLOS ONE.

For the revised submission, we would like to update our Funding Statement as follows: “This work is supported by National Natural Science Foundation of China under Grants No. 11575036 and No. 11805021, and BUPT Excellent Ph.D. Students Foundation under Grant No. CX2019138. The funders had no role in study design, data collection and analysis, decision to publish, or preparation of the manuscript.”

In the initial submission, we provided the funding statement in the manuscript but did not state them in the Funding section of the online submission. We must apologize for our carelessness. We would be grateful if you could help us update the financial disclosure statement. 

The followings are the details of the reply. 

Reviewer #1: The authors have addressed my concerns. Ambiguities have been removed and the English is now sufficiently intelligible.

Reviewer #2: The authors of the manuscript “Dynamics in the Sakaguchi-Kuramoto Model with bimodal frequency distribution” investigate the dynamics of a population of phase oscillators whose natural frequency terms are drawn from a bimodal distribution. The coupling between oscillators is global and occurs via the sine of their pairwise phase differences. A phase lag parameter, or frustration parameter, is added, which leads to a bimodal Kuramoto-Sakaguchi model. The model in this particular form and with respect to asymmetric frequency distributions has not been analyzed in the literature, yet. Moreover, the authors report the “revival” of the incoherent state: when increasing a bifurcation parameter, the stable incoherent state first becomes unstable and later on stabilizes again. The collective behavior of the system is studied following the Ott-Antonsen ansatz, which allows for deriving analytically tractable low-dimensional dynamics of the Kuramoto order parameter. Here, the system consists of three coupled ordinary differential equations, whose bifurcation structure is investigated with analytic and numerical techniques.

Comment 1: Although the here-reported form of the Kuramoto-Skaguchi model is new, the mathematically sound results for the symmetric case, which is central to this manuscript, can be anticipated from Martens, Bick and Panaggio (Chaos 26, 094819, 2016) and Refs. 20, 29, when taking into account that there is an equivalence between the bimodal formulation and a two-coupled-population formulation of coupled phase oscillators as established, e.g., in Ref. 30. The manuscript can present an important additional contribution to the field if, in a revised version, the authors can explicate whether the phase lag parameter \\beta and the heterogeneity parameter \\Delta have differential effects on the collective dynamics.

Reply: Thanks for the comment. Following the suggestions, we have made a brief discussion on the effects of \\Delta and \\beta on collective dynamics in the revised manuscript at lines 78-88 on page 3.

Comment 2: The novelty of this manuscript is the impact of the asymmetric bimodal distribution on the revival of the incoherent state, which unfortunately is not yet worked out thoroughly. Since similar revival results have been obtained by Omelchenko and Wolfrum in for unimodal frequency distributions, the manuscript can significantly be improved when the authors discuss in more detail the differences and similarities with their work and that by Omelchenko and Wolfrum.

Reply: Thank you very much for the suggestion. In the revised manuscript, we have discussed in more detail the differences and similarities with the work by Omelchenko and Wolfrum at lines 276-286 on pages 9.

Comment 3: As to the presentation of results, the authors should clarify why unstable solutions and their bifurcations (as in Figs. 1, 2, 3 and 4) are studied extensively. As their numerical analysis attains large attention through Figs. 1 and 2, the reader is misled in that unstable solutions seem to be crucial. But I doubt that they have any real effect on the dynamics of the population activity [apart from the SN2 curve in Fig. 3b between the CP and TB points].

Reply: We agree with that unstable solutions have no any effect on the long-term dynamics of the population dynamics. However, the existence of unstable solutions greatly shapes the topological structure of the underlying phase space and has strong impacts on the transient dynamics of the model. Moreover, these unstable solutions might become stable with the change of parameter and take effects on the long-term dynamics of the model. Therefore, in the perspective of stability analysis, the exploration of unstable solutions is necessary. We have explained it at lines 193-199 on pages 7.

Comment 4: The presentation of the resulting dynamical regimes appears unclear: why are the sub-order parameters r_1 and r_2 the important properties? The insets in Fig. 3 suggest that r_1 and r_2 really are the central properties, but why they are of interest, does not become clear. And why is not the global order parameter Z characteristic for the network dynamics?

Reply: Thanks for the comment. Actually, the network dynamics can be reflected by the global order parameter Z. If we just numerically simulate the coupled phase oscillators Eqs. (1,2), Z will be the candidate for exploring the population dynamics. However, in this work, we study the population dynamics by considering the low-dimensional reduced model, deduced based on the OA ansatz, in which the sub-order parameters r_1, r_2, and psi are the state variables. Consequently, we pay attention to r_1, r_2, and \\psi to explore the population dynamics. We have emphasized this point at the lines 112-114 on page 4.

Comment 5: The insets in Fig. 3 are hard to understand. A more intuitive explanation, and/or schematics of the global states, would be helpful for the reader. Maybe the layout of inset figures can be changed to that used in Bick, Martens and Panaggio (Chaos 2016, Chaos 2018).

Reply: Thanks for the comment. We have rewritten the figure caption in Fig. 3 and try to make it to be readable. We also follow the suggestions and have updated Fig. 3 by adding arrows to the trajectories of evolution. The solutions and trajectories on the plane of r_1 and r_2 could be distinguished. We hope the modifications are helpful for the reader. Besides, we have included the two related works in the references (Refs. [21], [34]). 

Comment 6: I don’t understand what Fig. 4 adds to the manuscript. If the purpose is to highlight how beta changes the bifurcation diagram, then there are at least two better options:

a. Add more bifurcation diagrams similar to Fig. 3 for different choices of beta.

b. Make Fig. 3b 3-dimensional by adding a beta-axis.

Reply: As we have worked out, the phase lag \\beta impacts on the dynamics in a non-monotonic way at large \\omega_0. For example, when \\beta increase from zero to \\pi/2, the incoherent state may be unstable for intermediate \\beta and stay stable otherwise. To figure out the non-monotomic effects of \\beta on the model dynamics, we presented the bifurcation diagrams on 4\\Delta/K-\\beta plane. 

To make our motivation clear, we have written the paragraph at lines 225-229 and 231-233 on pages 7 and 8 in the revised manuscript.

Comment 7: Line 220: Please start a new subsection in order to distinguish the parts using a symmetric versus an asymmetric distribution, respectively.

Reply: Thank you for the suggestion. We have modified the manuscript by adding new subsections on pages 4 and 8.

Comment 8: In Fig. 5c, \\omega_0 takes on negative values, but since it is considered to be half the distance between the peaks of the bimodal frequency distribution, how can it be negative? Please clarify.

Reply: Thanks for the comment. In the case of the asymmetrical bimodal frequency distribution where different peaks have different widths, \\omega_0 is defined as \\omega_1-\\omega_2 where \\omega_1 is center frequency of the peak with width \\Delta_1 while \\omega_2 for the peak with width \\Delta_2. Therefore, \\omega_0<0 means that the peak frequency \\omega_1<\\omega_2 while \\omega_0>0 means that the peak frequency \\omega_1>\\omega_2. These two situations are not the same. In this revised manuscript, we have clarified it at lines 252-255 on page 8.

Comment 9: In the Conclusion, line 257, please enumerate here your main findings in which way (how) beta affects the dynamics. Moreover, it would helpful to show whether beta has any differential effects compared to the other important parameter \\Delta because, in general, both prevent the system to fully synchronize. In line 211 the authors already point to the counter-intuitive phenomenon that close to 4\\Delta = K, a larger frustration parameter \\beta “tend[s] to destabilize the incoherent state”, but the mechanism why this is so remains unclear.

Reply: 

1.“In the Conclusion, line 257, please enumerate here your main findings in which way (how) beta affects the dynamics.” 

We have followed the suggestion and revised the conclusion at lines 294-297 on page 9.

2.“Moreover, it would helpful to show whether beta has any differential effects compared to the other important parameter \\Delta because, in general, both prevent the system to fully synchronize.” 

This comment is similar to comment 1, we have presented a comparison between \\beta and \\Delta on model dynamics at at lines 78-88 on page 3.

3.“In line 211 the authors already point to the counter-intuitive phenomenon that close to 4\\Delta = K, a larger frustration parameter \\beta “tend[s] to destabilize the incoherent state”, but the mechanism why this is so remains unclear.“ 

The comment is similar to comment 6. As presented in the reply to comment 6, we have rewritten the corresponding contend to make our points clear. 

Comment 10: Line 259: what are these conditions for better observing the revival? Please write them out here explicitly and, perhaps, the authors can also elaborate here on their hypothesis in lines 248/249.

Reply: Thank you very much for the suggestion. We have phrased the conditions for the revival of incoherent state at lines 276-286 on pages 9 in the revised manuscript.

---

## [Editor Report · Decision Letter 2]

18 Nov 2020

Dynamics in the Sakaguchi-Kuramoto Model with bimodal frequency distribution

PONE-D-20-12498R2

Dear Dr. Guo,

We’re pleased to inform you that your manuscript has been judged scientifically suitable for publication and will be formally accepted for publication once it meets all outstanding technical requirements.

Kind regards,

Per Sebastian Skardal

Academic Editor

PLOS ONE
---

## [Editor Report · Acceptance letter]

23 Nov 2020

PONE-D-20-12498R2 

Dynamics in the Sakaguchi-Kuramoto Model with bimodal frequency distribution 

Dear Dr. Guo:

I'm pleased to inform you that your manuscript has been deemed suitable for publication in PLOS ONE. Congratulations! Your manuscript is now with our production department. 

Kind regards, 

on behalf of

Dr. Per Sebastian Skardal 

Academic Editor

PLOS ONE